# A facile synthesis of *α,β*-unsaturated imines via palladium-catalyzed dehydrogenation

Chunyang Zhao[1,3], Rongwan Gao[2,3], Wenxuan Ma[1], Miao Li[1], Yifei Li[1], Qian Zhang [1], Wei Guan [2] ✉ & Junkai Fu [1] ✉

The dehydrogenation adjacent to an electron-withdrawing group provides an efficient access to *α,β*-unsaturated compounds that serving as versatile synthons in organic chemistry. However, the *α,β*-desaturation of aliphatic imines has hitherto proven to be challenging due to easy hydrolysis and preferential dimerization. Herein, by employing a pre-fluorination and palladium-catalyzed dehydrogenation reaction sequence, the abundant simple aliphatic amides are amendable to the rapid construction of complex molecular architectures to produce *α,β*-unsaturated imines. Mechanistic investigations reveal a Pd(0)/Pd(II) catalytic cycle involving oxidative H–F elimination of *N*-fluoroamide followed by a smooth *α,β*-desaturation of the in-situ generated aliphatic imine intermediate. This protocol exhibits excellent functional group tolerance, and even the carbonyl groups are compatible without any competing dehydrogenation, allowing for late-stage functionalization of complex bioactive molecules. The synthetic utility of this transformation has been further demonstrated by a diversity-oriented derivatization and a concise formal synthesis of (±)-alloyohimbane.

The introduction of an olefin motif adjacent to an electron-withdrawing group via C(*sp*³)–H dehydrogenation has received considerable attention in the last decades, giving rise to the corresponding *α,β*-unsaturated counterparts in an atom-economical manner[1–7]. The presence of multiple functionalities including electronically polarized double bond, electron-withdrawing group, and potential allylic C–H bonds demonstrates the synthetic utility and versatility of the *α,β*-unsaturated structural motifs in organic chemistry[8–11]. Significant advances have been successfully achieved in the *α,β*-dehydrogenation of aliphatic aldehydes/ketones[1,12–25], as well as less acidic nitriles[26,27], esters[27–31], amides[32–39], and carboxylic acids[40–45] through either direct or two-step protocols (Fig. 1a). In comparison, to the best of our knowledge, the *α,β*-dehydrogenation of aliphatic imines has not yet been developed. Despite the construction of *α,β*-unsaturated imines mainly relies on condensation of ene-aldehydes and amines in the current stage[46,47], the low-boiling points of simple ene-aldehydes and the tedious preparation

procedures for functionalized ene-aldehydes have partly limited its practicability[48–50]. Taking this in account, together with the widespread application of *α,β*-unsaturated imines in a number of C–C/N bond formations[51–54], searching for supplementary methods for the efficient synthesis of structurally diverse *α,β*-unsaturated imines, for example the dehydrogenation of abundant *N*-containing compounds, is highly desirable. However, several inherent challenges accompany with this dehydrogenation hypothesis. The imines, especially for the unstabilized aliphatic ones are sensitive to hydrolysis because of the kinetic equilibrium with the parent carbonyl compounds, which makes the purification and handling of these imines difficult. Moverover, the preference to dimerization is another major issue in the functionalization of aliphatic imines[55,56].

Actually, the research on dehydrogenation of *N*-containing compounds has been going on over the last decades[57]. Both stoichiometric and catalytic dehydrogenation of amines/amides has been extensively investigated to produce conjugated aryl imines or

¹Department of Chemistry, Jilin Province Key Laboratory of Organic Functional Molecular Design and Synthesis and Institute of Functional Material Chemistry, Northeast Normal University, Changchun 130024, China. ²Department of Chemistry, Institute of Functional Material Chemistry, Northeast Normal University, Changchun 130024, China. ³These authors contributed equally: Chunyang Zhao, Rongwan Gao. ✉e-mail: guanw580@nenu.edu.cn; fujk109@nenu.edu.cn

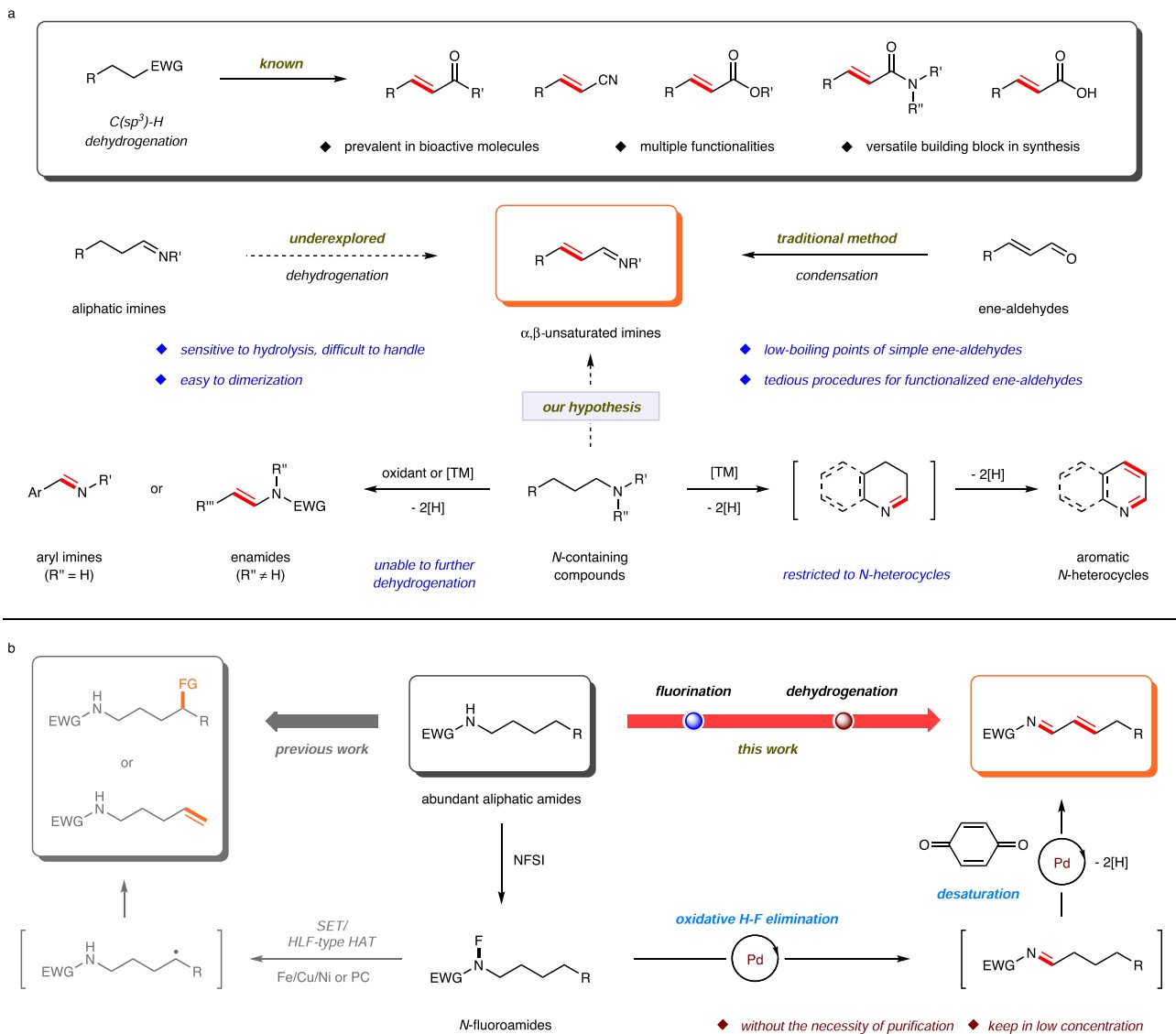

**Fig. 1 | Dehydrogenation strategies for the preparation of unsaturated compounds. a** α,β-Desaturation of electron-withdrawing groups, and the dehydrogenation of N-containing compounds. **b** The palladium-catalyzed dehydrogenation of amides to α,β-unsaturated imines (this work). EWG, electron-withdrawing group; NFSI, N-fluorobenzenesulfonimide; TM, transition metal; SET, single-electron transfer; HLF, Hofmann-Löffler-Freytag; HAT, hydrogen atom transfer.

tertiary enamides under mild reaction conditions[58–65], which, however, are unable to undergo further dehydrogenation. Meanwhile, this dehydrogenation can also act as an intermediate process in α-functionalization of amines/amides[66,67], and the in-situ formed aliphatic imines serve as transient intermediates in some intriguing reaction pathways rather than further dehydrogenation. In addition, the oxidative dehydrogenation of N-heterocycles provides an efficient access to functionalized aromatic N-heterocycles[68,69]. Mechanistic studies show that the presence of one or more N atoms can facilitate the reactions, which proceed via imine intermediate formation and a subsequent dehydrogenation. These literatures theoretically suggest the possibility of α,β-dehydrogenation of aliphatic imines, but also indicate a challenge due to the lack of aromatization as an additional driving force.

As an easily available and bench-stable N–F reagent, the N-fluoroamides have recently served as a particularly attractive synthon in organic synthesis (Fig. 1b). Upon treatment with transition metal (Fe, Cu, Ni) or photocatalysis, the N-fluoroamides can undergo N–F reduction via single-electron transfer (SET) followed by Hofmann-

Löffler-Freytag (HLF) type hydrogen atom transfer (HAT) to induce remote C($sp^3$)–H functionalization[70–80] or intramolecular cyclization[81]. Recently, Nagib and co-workers reported a dual photoredox- and copper-catalyzed remote C–H desaturation of N-fluoroamides to furnish δ vinyl amides, that could be converted to diverse families of medicinal motifs[82].

Herein, we disclose an interesting reaction model for N-fluoroamides, and employ palladium catalyst in the dehydrogenation of N-fluoro-sulfonamides to furnish α,β-unsaturated imines with good efficiency. The reaction proceeds via an initial Pd(0)-mediated oxidative H–F elimination and a subsequent Pd(II)-catalyzed desaturation process with smooth potential energy surface. The in-situ generation of aliphatic imine intermediate avoids the necessity of purification and prevents the accumulation of imine intermediate for further dimerization. It is remarkable that through a two-step strategy combining convenient pre-fluorination and this dehydrogenation reaction, the abundant simple aliphatic amides are allowed for the rapid construction of complex molecular architectures to forge versatile α,β-unsaturated imines.

## Results

### Screening of the reaction conditions

Exploration of the dehydrogenation reaction started with palladium-catalyzed conditions employing *N*-fluoro-sulfonamide **1** as the model substrate. As outlined in Table 1, with catalytic amount of Pd(OAc)$_2$ and bidentate ligand DPEPhos, *N*-fluoro-sulfonamide **1** could undergo the dehydrogenation in dioxane at 70 °C in the presence of BQ as oxidant and HOAc as additive, giving rise to the corresponding $\alpha,\beta$-unsaturated imine **2** in a good yield of 84% along with a small amount (5%) of defluorination by-product **3** (entry 1). The experimental operation was simple, and the reaction could even be undertaken exposing to air with a similar efficiency to that under N$_2$ atmosphere (entry 2). Brønsted acid facilitated the transformation but was not a necessity; without HOAc, the desired dehydrogenation product **2** was isolated in a slightly decreased yield of 74% (entry 3). Other bidentate phosphine ligands including Dppf and BINAP showed catalytic activity, however, they were far less efficient than DPEPhos. Moreover, without the addition of external ligand, no desired product was observed (entries 4–6). The palladium catalysis played a critical role in this reaction. The dehydrogenation reaction proved to be ineffective by replacing Pd(OAc)$_2$ with either Pd(PPh$_3$)$_2$Cl$_2$ or Pd(PPh$_3$)$_4$ (entries 7 and 8), while the use of Pd$_2$(dba)$_3$ led to a moderate yield of 36% (entry 9). Other types of

transition metals, such as Cu(OAc)$_2$, Fe(OTf)$_2$[70], and Ni(acac)$_2$[79] that frequently employed in the functionalization of *N*-fluoroamides have been tested, but no desired product **2** could be detected; instead, defluorination by-product **3** was obtained ranging from 40 to 16% yields along with the recovery of starting material **1** (entries 10-12).

It should be mentioned that *N*-fluoro-sulfonamide **1** itself could serve as the terminal oxidant; without the addition of BQ, $\alpha,\beta$-unsaturated imine **2**, and amide by-product **3** were obtained in 45 and 42% yields, respectively (entry 13). Thus, the employment of BQ tactfully prevented the unnecessary consumption of *N*-fluoro-sulfonamides. However, other oxidants, such as selectfluor, *m*CPBA, and LPO made negative effects on the reaction outcomes, offering the desired product **2** in very low yields (entries 14–16). We have also tried to employ O$_2$ as the terminal oxidant by performing the reaction with catalytic amount (0.2 equiv) of BQ under O$_2$ balloon. Unfortunately, it seemed that O$_2$ took a very limited participation in the reaction presumably due to the short reaction time that did not match the slow turnover between BQ and O$_2$ (entry 17). Switching the reaction solvent to THF (63%), DCE (27%) or DMSO (13%) gave inferior yields, and the yields of amide by-product **3** have increased accordingly (entries 18-20). Furthermore, some control experiments have been carried out. Either the acidic conditions excluding palladium catalysis or the basic conditions[83] by replacing HOAc with Et$_3$N or K$_2$CO$_3$ were unable to

**Table 1 | Selected Optimization Results**[a,b]

| Entry | Deviation from standard conditions | Yield of 2 (%) | Yield of 3 (%) |
|---|---|---|---|
| 1 | None | 84 (74%)[d] | 5 |
| 2 | under N$_2$ | 85 | 7 |
| 3 | no HOAc | 74 | 10 |
| 4 | Dppf instead of DPEPhos | 51 | 7 |
| 5 | BINAP instead of DPEPhos | 21 | 3 |
| 6 | no DPEPhos | 0 | 4 |
| 7 | Pd(PPh$_3$)$_2$Cl$_2$ instead of Pd(OAc)$_2$ | 0 | 2 |
| 8 | Pd(PPh$_3$)$_4$ instead of Pd(OAc)$_2$ | 0 | 25 |
| 9 | Pd$_2$(dba)$_3$ instead of Pd(OAc)$_2$ | 36 | 6 |
| 10 | Cu(OAc)$_2$ instead of Pd(OAc)$_2$ | 0 | 40 (10 of **1**)[c] |
| 11 | Fe(OTf)$_2$ instead of Pd(OAc)$_2$ | 0 | 32 (40 of **1**)[c] |
| 12 | Ni(acac)$_2$ instead of Pd(OAc)$_2$ | 0 | 16 (68 of **1**)[c] |
| 13 | no BQ | 45 | 42 |
| 14 | selectfluor instead of BQ | 8 | 16 |
| 15 | *m*CPBA instead of BQ | 5 | 43 |
| 16 | LPO instead of BQ | 5 | 13 |
| 17 | BQ (0.2 equiv) under O$_2$ atmosphere | 58 | 30 |
| 18 | THF instead of dioxane | 63 | 8 |
| 19 | DCE instead of dioxane | 27 | 17 |
| 20 | DMSO instead of dioxane | 13 | 32 |
| 21 | no Pd(OAc)$_2$ | 0 | 0 (85 of **1**)[c] |
| 22 | Et$_3$N or K$_2$CO$_3$ (1.0 equiv) instead of HOAc | 0 | 0 (70 or 63 of **1**)[c] |

*BQ* 1,4-benzoquinone, *DPEPhos* bis[2-(diphenylphosphino)phenyl]ether, *Dppf* 1,1'-bis(diphenylphosphino)ferrocene, *BINAP* 1.1'-binaphthyl-2.2'-diphemylphosphine, *dba* dibenzylideneacetone, *m*CPBA 3-chloroperoxybenzoic acid, *LPO* lauroyl peroxide.
[a]Standard reaction conditions: a mixture of **1** (0.20 mmol), Pd(OAc)$_2$ (0.02 mmol), DPEPhos (0.022 mmol), HOAc (0.20 mmol), and BQ (0.20 mmol) in dioxane (1.0 mL) was stirred under air at 70 °C for 15 min.
[b]Isolated yields.
[c]The recovery of starting material **1**.
[d]The yield in a 5.0 mmol scale.

initiate the reaction with the primary mass balance being the unreacted starting material **1** (entries 21 and 22).

## Substrate scope

With the optimized conditions in hand, the substrate scope for the dehydrogenation reaction was investigated. As shown in Fig. 2, the linear *N*-fluoro-sulfonamides with either acyclic or cyclic alkyl chains could all be converted to the corresponding α,β-unsaturated imines **4**–**13** in good yields. In addition to the desired α,β-unsaturated imines, defluorinated aliphatic amides were the main by-products, together with some unidentified hydrolysis by-products. Besides different carbon scaffolds, a wide variety of functional groups were well tolerated. The reaction of *N*-fluoro-sulfonamide bearing a remote phenyl group provided the desired product **14** in 81% yield. For the unsaturated carbon–carbon bonds, both terminal/internal alkenyl and alkynyl moieties could survive to produce **15**–**18**, wherein a minor *Z* isomer of the C−N double bond was observed for product **17** (*E/Z* = 9:1). The substrates bearing ester (**19**), amide (**20**),

**Fig. 2 | Substrate scope.** Standard conditions: a mixture of *N*-fluoro-sulfonamide (0.20 mmol), Pd(OAc)₂ (0.02 mmol), DPEPhos (0.022 mmol), HOAc (0.20 mmol), and BQ (0.20 mmol) in dioxane (1.0 mL) was stirred under air at 70 °C for 15 min. Isolated yields are shown. [a]*E/Z* ratio of the C−N double bond. [b]At 50 °C for 0.5 h. [c]For 0.5 h. DPEPhos, bis[2-(diphenylphosphino)phenyl]ether; BQ, 1,4-benzoquinone; Ts, tosyl; Bz, benzoxyl; Bn, benzyl; TBS, *t*-butyldimethylsilyl.

**41**, 75%
from lithocholic acid

**42**, 67% (E/Z = 9:1)[a]
from linoleic acid

**43**, 72% (E/Z = 9:1)[a]
from ketoprofen

**44**, 68% (E/Z = 19:1)[a]
from Ibuprofen

**45**, 73% (E/Z = 19:1)[a]
from stigmasterol

**46**, 67%
from celecoxib

**Fig. 3 | The dehydrogenation of *N*-fluoro-sulfonamides derived from bioactive molecules.** Standard conditions: a mixture of *N*-fluoro-sulfonamide (0.20 mmol), Pd(OAc)$_2$ (0.02 mmol), DPEPhos (0.022 mmol), HOAc (0.20 mmol), and BQ (0.20 mmol) in dioxane (1.0 mL) was stirred under air at 70 °C for 15 min. Isolated yields are shown. [a]*E/Z* ratio of the C–N double bond.

ether (**21**), or silyl ether (**22**) groups were tested, delivering the desired products in good yields. Moreover, the existence of halide atoms did not affect the dehydrogenation reaction (**23**, **24**), and the remaining bromo or chloro atoms could act as potential handles for further transformations. It should be mentioned that the reaction could be selectively carried out in the presence of carbonyl moieties (**25–27**), and no competing α,β-unsaturated carbonyls were observed. Then the branched *N*-fluoro-sulfonamides were evaluated. When *N*-fluoro-sulfonamides containing a tertiary carbon center at C4 position were employed as the substrates, the corresponding β,β-disubstituted α,β-unsaturated imines **28** and **29** were obtained in 65 and 71% yields, respectively. Moreover, this reaction was compatible with the substrates bearing alkyl side chains at C3 position. The desired α,β-disubstituted α,β-unsaturated imines **30** and **31** were obtained in acceptable yields, while the regioisomer **32b** with internal olefin moiety dominated the product distribution during the formation of **32**. In addition, the electronically diverse substituents on the phenyl ring of arylsulfonamide group including neutral (–Me), electron-donating (–OMe), and electron-withdrawing (–F, –CF$_3$, –NO$_2$) substituents had little impact on the dehydrogenation reaction to furnish the corresponding products **33–39** in good yields. Further treatment of *N*-fluoro-2-naphthalene-sulfonamides with the optimized reaction conditions afforded α,β-unsaturated imine **40** in 70% yields. Unfortunately, the *N*-fluoro-sulfonamides derived from secondary and cyclic amides failed to participate in the dehydrogenation reaction, with the majority of the substrates recovered.

In an effort to show the synthetic potential of this methodology in late-stage functionalization of biologically relevant molecules, several derivatives of natural products and pharmaceuticals were subjected to the optimized reaction conditions (Fig. 3). We were delighted to find that the *N*-fluoro-sulfonamides prepared from lithocholic acid (**41**), linoleic acid (**42**), ketoprofen (**43**), Ibuprofen (**44**), stigmasterol (**45**), and celecoxib (**46**) could all smoothly undergo the dehydrogenation reaction to afford the desired α,β-unsaturated imines in good yields ranging from 75 to 67%.

### Derivatization and a formal synthesis of (±)-alloyohimbane

The utility of this transformation for further assembly of valuable nitrogen-containing molecules was shown in Fig. 4. A selective 1,2-addition of α,β-unsaturated imine **2** could be achieved by H−, C(*sp*)−, or C(*sp*$^2$)−based nucleophiles to produce synthetically valuable allyic amides **47–49** in excellent yields[84]. A greater variety of structurally diverse allyic amides **50–52** were obtained by using C(*sp*$^3$)−based nucleophiles. When benzyl Grignard reagent was utilized as the nucleophile, enamide **53** was formed in 76% yield via an 1,4-addition of compound **2**. Moreover, the α,β-unsaturated imine could serve as a suitable aza-diene to take part in hetero-Diels−Alder reaction to efficiently produce tetrahydropyridine derivatives **54** and **55** in a highly regio- and diastereoselective manner.

Then a concise formal synthesis of (±)-alloyohimbane was conducted. Alloyohimbane is a member of the rauwolfia alkaloid family featuring a characteristic pentacyclic indole ring framework. Due to its complex structure and interesting biological activity, the synthesis of alloyohimbane has piqued the interest of organic chemists for decades[85,86]. Our synthesis commenced with α,β-unsaturated imine **31**. An intermolecular hetero-Diels−Alder reaction using 1,1-dimethoxyethene **56** as the dienophile followed by hydrolysis generated lactam **57** in 63% overall yield. Hydrogenation of the olefin moiety and a subsequent reductive removal of the tosyl group resulted in the formation of *cis*-octahydroisoquinolin-3-one **58**. Nucleophilic substitution reaction between unprotected amide **58** and mesylate **59** in the presence of NaH as the base furnished adduct **60**. After removal of the *tert*-butoxycarbonyl group under acidic conditions, the resulting compound **61** could be further converted to alloyohimbane via Bischler-Napieralski closure of the C-ring[87–90].

### Mechanistic studies

To shed light on the reaction mechanism, several control experiments were carried out as shown in Fig. 5. Performing the dehydrogenation reaction of *N*-fluoro-sulfonamide **1** in the presence of either 2,2,6,6-tetramethylpiperidinooxy (TEMPO) or butylated hydroxytoluene (BHT) as a radical scavenger slightly decreased the reaction efficiency, excluding the possibility of a radical pathway (Fig. 5a). When freshly prepared aliphatic imine **62** was subjected to the standard reaction conditions, the desired dehydrogenation product **5** was obtained despite in a low yield of 6%; in contrast, the by-products **63** and **64**, dimerized from imine and hydrolyzed aldehyde, respectively[55,56,83], dominated the reaction outcome along with some unidentified hydrolysis by-products. Carrying out the reaction under standard conditions by slowly adding imine **62** in dioxane (1.0 mL) using syringe pump, the yield of α,β-unsaturated imine **5** has increased to 13%, but the dimer by-product **63** (25% yield, 50% yield based on **62**) still dominated the reaction outcome, suggesting a very fast dimerization process. Interestingly, the yield of product **5** was improved to 17% when the concentration of reaction

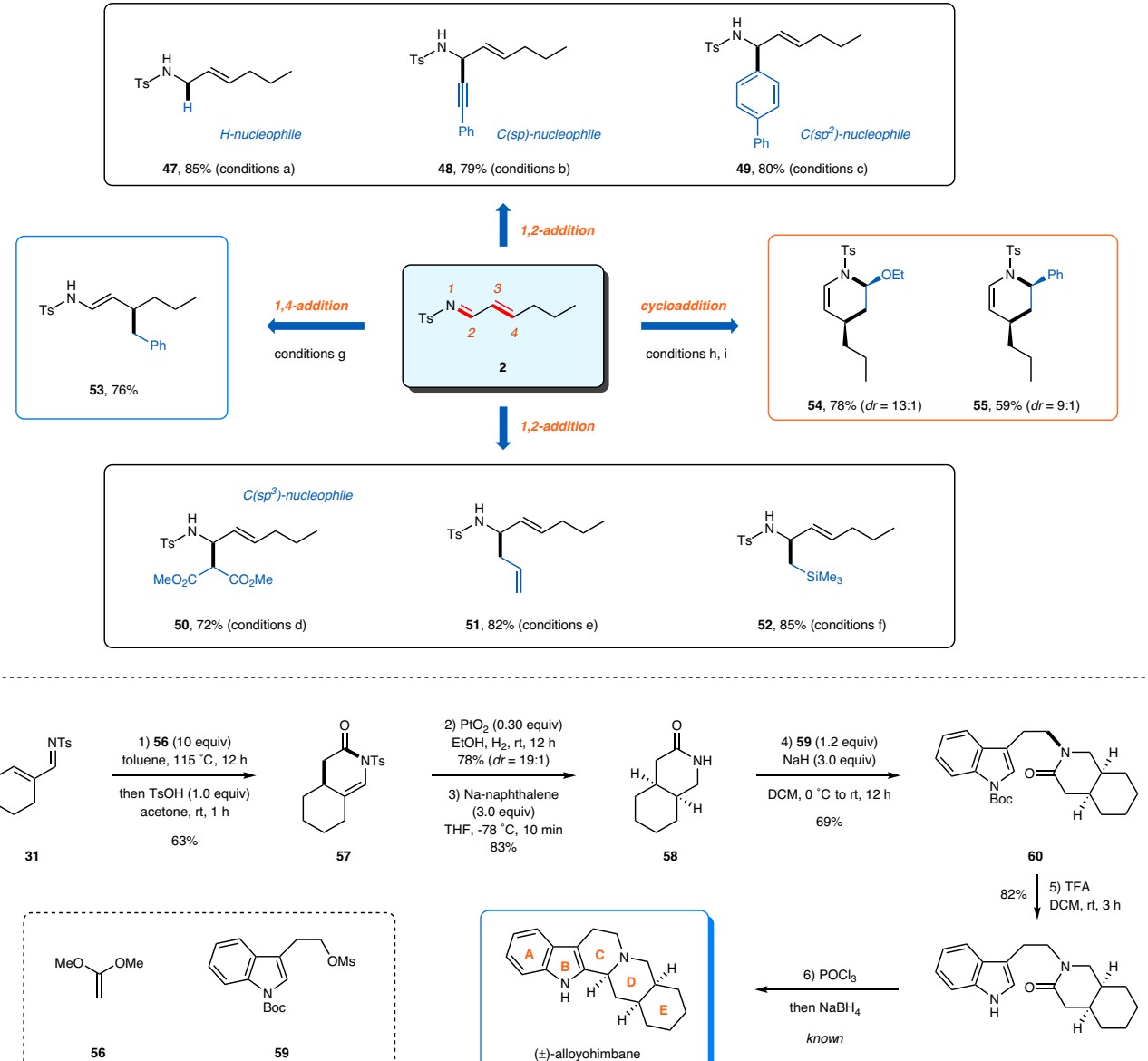

**Fig. 4 | Representative derivatization and a formal synthesis of (±)-alloyo-himbane.** Conditions (**a**) **2** (0.20 mmol), diisobutylalumium hydride (0.30 mmol), THF, −78 °C to rt, 12 h; (**b**) **2** (0.20 mmol), (phenylethynyl)magnesium bromide (0.60 mmol), Et$_2$O, −20 °C to rt, 12 h; (**c**) **2** (0.20 mmol), [1,1′-biphenyl]−4-ylmag-nesium bromide (0.60 mmol), Et$_2$O, −20 °C to rt, 12 h; (**d**) **2** (0.20 mmol), dimethyl malonate (0.60 mmol), La(OTf)$_3$ (0.02 mmol), Ph-PyBox (0.02 mmol), 4 Å MS, DCM, 60 °C, 12 h; (**e**) **2** (0.20 mmol), allylmagnesium chloride (0.60 mmol), Et$_2$O, −20 °C to rt, 12 h; (**f**) **2** (0.20 mmol), [(trimethylsilyl)methyl]magnesium chloride (0.60 mmol), Et$_2$O, −20 °C to rt, 12 h; (**g**) **2** (0.20 mmol), benzylmagnesium chloride (0.60 mmol), Et$_2$O, −20 °C to rt, 12 h; (**h**) **2** (0.20 mmol), ethoxyethene (2.0 mmol), 2,2,2-trifluoroethanol, 50 °C, 24 h; (**i**) **2** (0.20 mmol), styrene (2.0 mmol), hexa-fluoroisopropanol, 110 °C, 24 h. TFA, trifluoroacetic acid; Boc, *t*-Butyloxycarbonyl; Ms, methanesulfonyl.

solution was diluted to 0.02 M, and the yields of dimer by-products **63** and **64** have decreased accordingly. Unfortunately, further diluting the reaction mixture would give inferior yields of product **5** (15% in 0.01 M, 9% in 0.005 M), probably due to a slow dehy-drogenation process under low concentration of palladium catalyst and BQ. These experimental results suggested that the dehy-drogenation reaction might proceed via imine intermediate, and a low concentration of the imine intermediate benefitted the avoid-ance of its dimerization (Fig. 5b). Then a series of deuterium labeling experiments were conducted to explore the possible pathway for the C–C double bond formation. Treatment of *N*-fluoro-sulfonamide **1** under standard conditions by replacing HOAc with DOAc resulted in the formation of **D-2**, wherein 15% deuterium was introduced on C3

position. Subjecting *N*-fluoro-sulfonamide **65** with C3 position blocked by two methyl groups to the standard reaction conditions in the presence of DOAc led to imine product **66** in 61% yield, and no deuterium was incorporated[91]. Moreover, the dehydrogenation reaction of **D-4s** deuterated at C3 position furnished the desired product **D-4** with 20% loss of the deuterium. Taken together, the C–C double bond formation via α-metallization of imine inter-mediate followed by β-H elimination is preferred over imine-directed β-C–H activation pathway (Fig. 5c)[43]. To further verify this assumption, enamide **53** was directly treated with the standard reaction conditions. As expected, α,β-unsaturated imine **67** was obtained in 60% yield without the observation of dimer by-product, indicating that enamide might serve as an intermediate in the

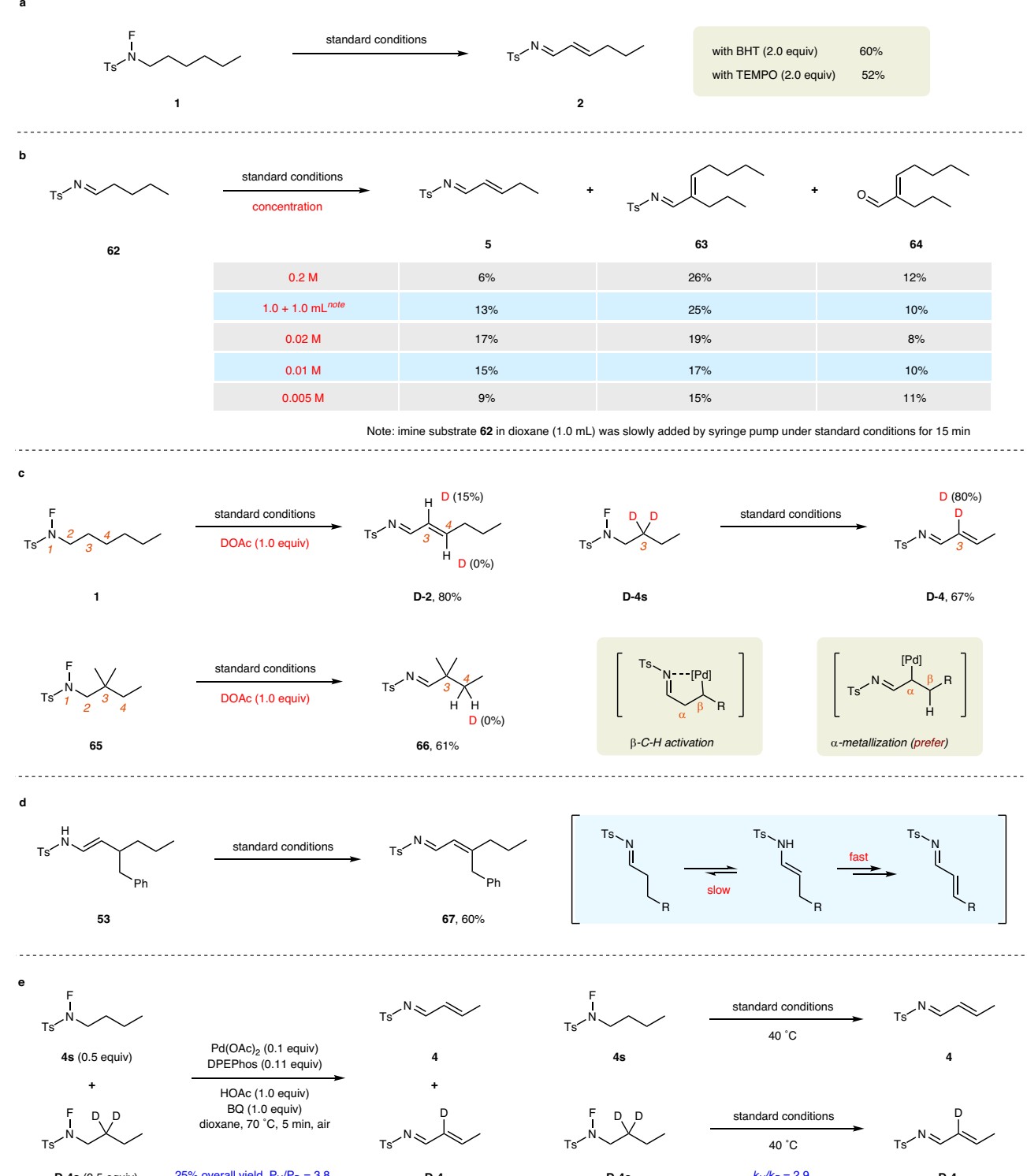

**Fig. 5 | Control experiments. a** Radical trapping experiments. **b** Direct dehydrogenation reaction of imine. **c** Deuterium labeling experiments. **d** Direct dehydrogenation reaction of enamide. **e** Kinetic isotope effect experiments. BHT, butylated hydroxytoluene; TEMPO, 2,2,6,6-tetramethylpiperidinooxy.

dehydrogenation reaction and the downstream process from enamide was much faster than the equilibrium between enamide and imine intermediate (Fig. 5d). What's more, evident primary kinetic isotope effect (KIE) values were observed ($P_H/P_D = 3.8$ in intermolecular competition and $k_H/k_D = 2.9$ in parallel reaction), indicating that the C–H bond-cleavage process may be involved in the rate-determining step (Fig. 5e).

## Theoretical calculations

To gain a deeper understanding of this palladium-catalyzed dehydrogenation of *N*-fluoro-sulfonamides to prepare $\alpha,\beta$-unsaturated imines, a Pd(0)/Pd(II) catalytic cycle involving oxidative H–F elimination, tautomerism, $\alpha$-metallization of imine and $\beta$-hydrogen elimination was proposed based on the aforementioned experimental evidences. The corresponding reaction mechanism was evaluated

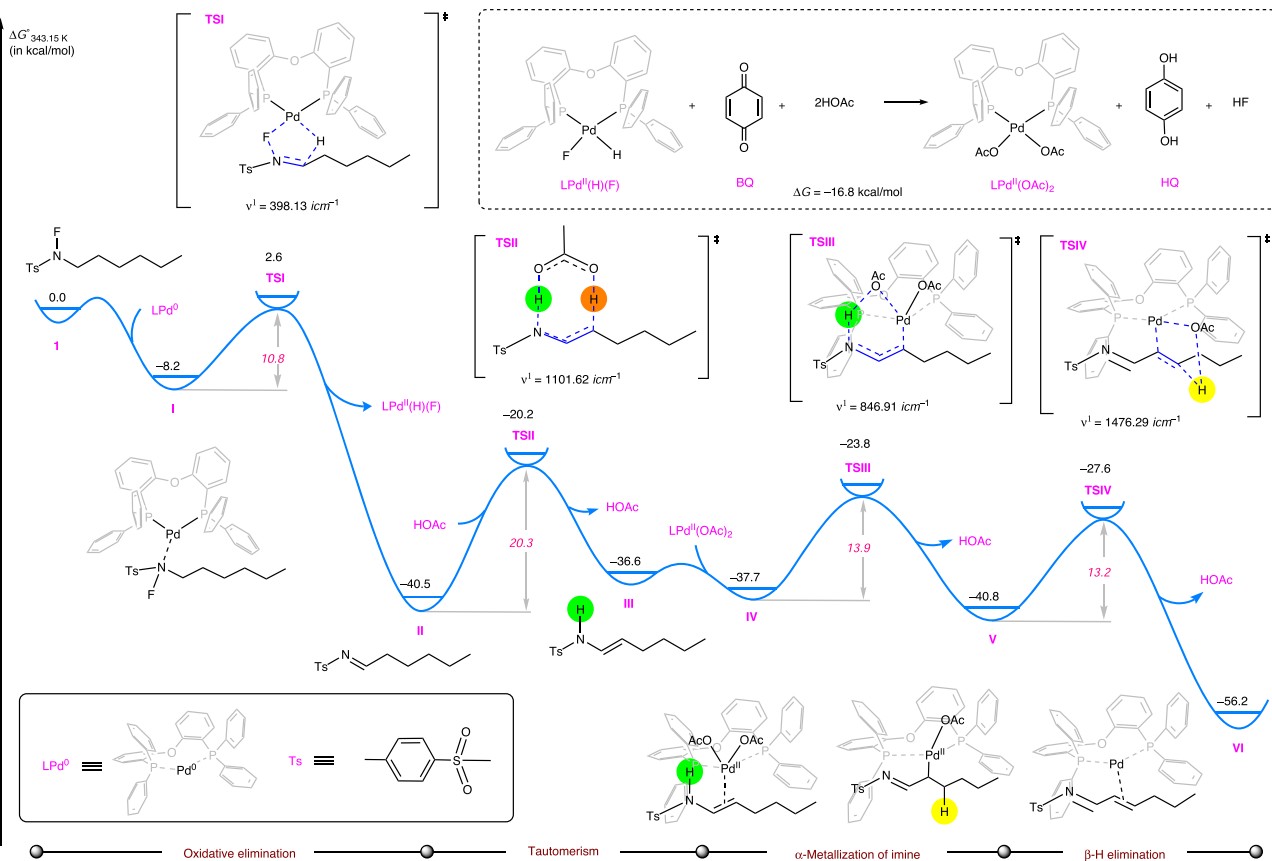

**Fig. 6 | Density functional theory calculations.** Gibbs energy profile ($\Delta G°$) of the Pd(0)/Pd(II) catalytic cycle.

through density functional theory (DFT) calculations at the SMD[92](dioxane)/M06[93]/[6-311 + + G(d,p)/SDD[94](Pd)]//M06/[6-31 G(d)/Lan L2DZ[95](Pd)] level (Fig. 6, see Part 2 in the Supplementary Information for computational details).

As a well-established process, it can be understood here that catalytically active Pd[0] species can be easily generated in situ via reduction of Pd[II](OAc)$_2$ catalyst precursor by bidentate phosphine ligand DPEPhos (L), wherein the *ortho*-oxygen atom would benefit the participation of electron-rich arylphosphine in the reduction[96,97]. As depicted in Fig. 6, the coordination of LPd[0] and *N*-fluoro-sulfonamide **1** forms a Pd(0) intermediate **I** with the Gibbs free energy change ($\Delta G°$) of −8.2 kcal/mol. Subsequently, a rare oxidative elimination involving H–F elimination and palladium oxidation occurs via a concerted five-membered-ring (Pd–F–N–C–H) transition state **TSI** with a small Gibbs activation energy ($\Delta G°‡$) of 10.8 kcal/mol and a negative $\Delta G°$ value of −32.3 kcal/mol to generate imine intermediate **II**. In contrast, the stepwise oxidative addition followed by *β*-H elimination is much less favorable than the present concerted pathway by 25.2 kcal/mol (Supplementary Fig. 1). The LPd[II](H)(F) complex has been successfully detected by both [1]H and [19]F NMR (for details, see Part 3.4 in the Supplementary Information). With the assistance of additional HOAc, the resultant imine intermediate **II** undergoes a proton transfer to isomerize into an enamine intermediate **III** through an eight-membered-ring transition state **TSII**. Such tautomerism is endoergic of 3.9 kcal/mol and requires a moderate energy barrier of 20.3 kcal/mol. In comparison, the Brønsted acid-free and water-assisted tautomerisms were evaluated to be unfavorable (Supplementary Fig. 2). In parallel to the tautomerism from imine to enamine (**II → III**), the Pd(II) hydrofluoride LPd[II](H)(F) generated by the initial oxidative elimination can be easily oxidized by BQ with the participation of HOAc[98]. Such spontaneous oxidation process of hydride is exoergic releasing

16.8 kcal/mol. The resultant species LPd[II](OAc)$_2$ weakly coordinates with the C = C double bond of **III** to form the Pd(II) complex **IV** with a slightly negative $\Delta G°$ value of −1.1 kcal/mol. Subsequently, *α*-metallization of imine in **IV** occurs via a six-membered-ring transition state **TSIII** to form the Pd(II)-alkyl intermediate **V** and release HOAc with a small $\Delta G°‡$ value of 13.9 kcal/mol and a negative $\Delta G°$ value of −3.1 kcal/mol, respectively. Such synergistic *α*-metallization process of imine is in line with the above deuterium labeling experiments (see Fig. 5c) as well as the control experiment with enamide (see Fig. 5d). Finally, the desired *α,β*-unsaturated imine product can be delivered by the *β*-H elimination of **V**. The five-membered-ring **TSIV** is the corresponding transition state for this step with a small $\Delta G°‡$ value of 13.2 kcal/mol.

Overall, the tautomerism from imine to enamine is the rate-determining step in accord with the KIE studies in Fig. 5e, and a smooth reaction potential energy surface allows for a rapid consumption of the in-situ generated aliphatic imine intermediate, which eventually keep the imine intermediate at low concentration to prevent its accumulation for further dimerization. This interpretation is in accord with the experimental results observed in Fig. 5b.

## Discussion

In summary, we utilize palladium catalyst in the functionalization of *N*-fluoroamides, and disclose a dehydrogenation reaction for the preparation of *α,β*-unsaturated imines. This protocol features broad substrate scope, simple experimental operation in air, high efficiency, and excellent functional group tolerance towards even the competing carbonyl groups. The successful application to biologically relevant molecules, diversity-oriented derivatization of the resultant *α,β*-unsaturated imines, and a concise formal synthesis of (±)-alloyohimbane fully demonstrate the great potential of this methodology in pharmaceutical industry. It should be mentioned that by combining a

simple pre-fluorination and the current dehydrogenation reaction, the abundant simple aliphatic amides are allowed to efficiently forge synthetically versatile α,β-unsaturated imines without the carbon skeleton rearrangement. Control experiments and DFT calculations reveal that the reaction proceeds via a Pd(0)/Pd(II) catalytic cycle involving oxidative H−F elimination of *N*-fluoroamide followed by α,β-desaturation of the resulting aliphatic imine intermediate, and a smooth reaction potential energy surface effectively avoids the accumulation of imine intermediate to dimerization. Further applications of this methodology in the synthesis of complex natural products and detailed mechanistic studies are in progress in our laboratory.

## Methods

### The general procedure A

To a dry Schlenk flask were added *N*-fluoro-sulfonamide **1** (54.6 mg, 0.20 mmol, 1.0 equiv), Pd(OAc)$_2$ (4.5 mg, 0.02 mmol, 0.10 equiv), bis[2-(diphenylphosphino)phenyl]ether (DPEPhos, 11.8 mg, 0.022 mmol, 0.11 equiv), 1,4-benzoquinone (BQ, 21.6 mg, 0.20 mmol, 1.0 equiv), anhydrous 1,4-dioxane (1.0 mL), and HOAc (11 µL, 0.20 mmol, 1.0 equiv). The mixture was stirred at 70 °C (oil bath) for 15 min under air. Once completion, the reaction was cooled to room temperature. The reaction mixture was filtered by celite, and the filtrate was concentrated *in vacuo*. Further purification by a flash column chromatography using eluents (PE/EA = 20:1) afforded the desired product **2** as colorless oil (42.2 mg, 0.17 mmol, 84%, *E/Z* > 20:1).

### The general procedure B

To a dry Schlenk flask were added *N*-fluoro-sulfonamide **18s** (71.8 mg, 0.20 mmol, 1.0 equiv), Pd(OAc)$_2$ (4.5 mg, 0.02 mmol, 0.10 equiv), DPEPhos (11.8 mg, 0.022 mmol, 0.11 equiv), BQ (21.6 mg, 0.20 mmol, 1.0 equiv), anhydrous 1,4-dioxane (1.0 mL), and HOAc (11 µL, 0.20 mmol, 1.0 equiv). The mixture was stirred at 50 °C (oil bath) for 30 min under air. Once completion, the reaction was cooled to room temperature. The reaction mixture was filtered by celite, and the filtrate was concentrated *in vacuo*. Further purification by a flash column chromatography using eluents (PE/EA = 20:1) afforded the desired product **18** as yellow oil (36.4 mg, 0.11 mmol, 54%, *E/Z* > 20:1).

Precaution: It would be better to finish the purification within one hour to avoid the product decomposition.

## Data availability

The authors declare that all data supporting the findings of this research are available within the article and its Supplementary Information. Cartesian coordinates of the calculated structures are available from the Source Data 1. Any further relevant data are available from the corresponding authors on request. Source data are provided with this paper.

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

## Acknowledgements

We gratefully acknowledge the National Natural Science Foundation of China (22371036, for J.F.; 21971034, for J.F.) and Jilin Province Scientific and Technological Development Program (20230508107RC, for J.F.) for financial support.

## Author contributions

C.Z. designed and performed the experiments. R.G. and W.G. performed the density functional theory calculations. W.M. and M.L. assisted in completing the experiments. Y.L. and Q.Z. analyzed the data. J.F. directed the project and wrote the manuscript. All the authors were involved in interpretation of the results presented in the manuscript.

## Competing interests

The authors declare no competing interests.
