## [Peer Review File · Nature Communications]

A Facile Synthesis of α,β -Unsaturated Imines via Palladium-Catalyzed DehydrogenationREVIEWER COMMENTS

Reviewer #1 (Remarks to the Author):

The authors describe a two step conversion of amines to unsaturated imines by two different oxidation reactions.

[Note from the Editor: Reviewer #1 suggested the authors to remove parts a and b of Figure 1 and describe the contribution of the present submission as accurately as possible.]

Reviewer #2 (Remarks to the Author):

This manuscript deals with the synthesis of α,β -unsaturated imines via fluorination of aliphatic amides followed by palladium-catalyzed dehydrogenation. This reaction is novel and original, and can be applied to amides with various functional groups, including those derived from bioactive molecules. The products are obtained in good to high yields. Furthermore, derivatization of the synthesized α,β -unsaturated imine to a variety of compounds and formal synthesis of alloyohimbane are also demonstrated. Thus, the present method is obviously synthetically useful. The reaction mechanism appears to involve rare and interesting steps such as oxidative H-F elimination and is supported by several control experiments and DFT calculations. The results obtained will attract readers not only in organometallic and synthetic organic chemistry, but also in related fields.

The approach is reasonable. The text is written in a manner that is easily understood by a wide range of readers, and the quality of presentation is high. The quality of the data is satisfactory and the experimental procedures are detailed enough to be reproduced. The interpretation of the data and conclusions are considered reasonable and reliable. References are adequate. The abstract is clearly written, and the introduction and conclusions are clearly and appropriately stated. Overall, the paper is considered acceptable for publication in Nature Communications after some revisions. Questions and comments are listed below.

1. Although the structure of DPEPhos is shown in Table 2, it should be moved to Table 1.
2. In Table 1, entry 13, the reaction proceeds even in the absence of BQ, yielding the product in 45%. How can this result be explained? According to Figure 4, BQ is involved in the regeneration of $LPd(OAc)_2$.
3. In Figure 4, the coordination bond of the alkene moiety to Pd in IV and VI should be drawn from the center of the alkene toward Pd.
4. In Figure 3b, the authors show the effect of imine concentration on the product yield. I recommend that the authors perform this reaction using a syringe pump to slowly add the imine. This would keep the concentration of the imine low throughout the reaction. Under these conditions, the desired compound 5 may be obtained in a higher yield than those shown in Figure 3b.
5. Have the authors attempted KIE (kinetic isotope effect) experiments using AcOD and/or compound D-4s? If the tautomerism from imine to enamine is the turnover-limiting step, as the authors have shown in Figure 4 based on the DFT calculations, KIE should be seen when the rates are compared between the reactions with AcOH or AcOD and compound 1 as a substrate, with AcOH and compound 4s (no D incorporated substrate of D-4s) or D-4s as a substrate, and with AcOH+4s or AcOD+D-4s. These data would further support the

proposed mechanism.

6. Can the authors reasonably explain why DPEPhos is more effective than other ligands?

7. Have the authors attempted to observe LPd(H)(F) produced by the oxidative H-F elimination? The complex may be observable by ^1H and ^{19}F NMR by a stoichiometric reaction at low temperature.

Reviewer #3 (Remarks to the Author):

Fu and co-workers describe a facile synthesis of α,β -unsaturated imines via palladium-catalyzed dehydrogenation. And this methodology has been applied to derivatization of biologically relevant molecules and a formal synthesis of (\pm)-alloyohimbane. However, this manuscript does not essentially solve the dehydrogenation of aliphatic imines, and it's just for the preparation of α,β -unsaturated imines from N-fluoroamide prepared with tedious multistep procedures. In fact, the challenges of preparation of α,β -unsaturated imines from imines were still not solved, such as easy hydrolysis and preferential dimerization. In addition, the direct α,β -dehydrogenation of imines have already been reported more than a decade ago, and it subsequently serves as an intermediate to take place other transformations (Adv. Synth. Catal. 2009, 351, 1229-1232; Chem. Asian J. 2009, 4, 1712-1716; Angew. Chem. Int. Ed. 2011, 50, 3920-3924). From the perspective of publication, I would recommend publication after address the following points.

1) The authors elaborate the developments in α,β -dehydrogenation of carbonyl compounds in the introduction in detail, and related references have been cited. However, relevant references have been missed, such as J. Org. Chem. 2019, 84, 8267-8274; Org. Lett. 2021, 23, 1611-1615; Org. Lett. 2023, 25, 4429-4433; Org. Biomol. Chem. 2017, 15, 7317-7320; Angew. Chem. Int. Ed. 2018, 57, 16205-16209; Adv. Synth. Catal. 2018, 360, 4774-4783. And they should be cited under the corresponding topics.

2) As the authors state, aliphatic imines are easy hydrolysis. But various α,β -unsaturated imines are isolated via a flash column chromatography in this manuscript. Precautions need to be listed in the Supplementary Information. The yields range from decent to moderate, but there is no discussion on the reasons for not having "100% yield" in these reactions. What are the side products? Hydrolysis products? Sequential dehydrogenative products (γ,δ -dehydrogenative products)?

3) Regarding the substrate scope, except for benzene sulfonyl groups on nitrogen, are other substituents feasible? Such as phenyl, alkyl? In the introduction, the author stated the oxidative dehydrogenation of N-heterocycles provides an efficient access to functionalized aromatic N-heterocycles. But dehydrogenation of N-heterocycles hadn't been exhibited in this manuscript. Maybe benzene sulfonyl groups on nitrogen are crucial. However, what are the advantages of this method compared with condensation between aldehydes and p-toluenesulfonamide? And the preparation of raw materials is complicated for this method. And a gram-scale should be presented to prove the practicability of the method rather than 0.20 mmol scale.

4) About reaction mechanism, a KIE experiment should be implemented to confirm the rate-determining step. The author stated a low concentration of the imine intermediate benefitted the avoidance of its dimerization (Figure 3b). However, only two sets of data are given. To better illustrate the tendency, more data should be given. Apart from DFT calculations, is there any other evidence about a concerted five-membered-ring (Pd-F-N-C-H) transition state TSI rather than the stepwise oxidative addition followed by β -H elimination, such as kinetic study?

5) Minor points: All " α, β " in the manuscript should be italicized. The line 282, "reveale" to "reveal".

Response to the Reviewer 1's comments

Comments:

The authors describe a two step conversion of amines to unsaturated imines by two different oxidation reactions.

[Note from the Editor: Reviewer #1 suggested the authors to remove parts a and b of Figure 1 and describe the contribution of the present submission as accurately as possible.]

Response: We thank the Reviewer for the positive evaluation and strong support of this work.

According to the valuable suggestion of the Reviewer, Figure 1 in the main text has been revised to describe our contribution on the construction of α,β -unsaturated imines in more detail.

Previous Figure 1:

Figure 1. Dehydrogenation strategies for the preparation of unsaturated compounds. a, α,β -Desaturation of electron-withdrawing groups. b, Dehydrogenation of *N*-containing compounds. c, The palladium-catalyzed dehydrogenation of amides to α,β -unsaturated imines (this work).

Revised Figure 1:

Response to the Reviewer 2's comments

Comments:

This manuscript deals with the synthesis of α,β -unsaturated imines via fluorination of aliphatic amides followed by palladium-catalyzed dehydrogenation. This reaction is novel and original, and can be applied to amides with various functional groups, including those derived from bioactive molecules. The products are obtained in good to high yields. Furthermore, derivatization of the synthesized α,β -unsaturated imine to a variety of compounds and formal synthesis of alloydhimbane are also demonstrated. Thus, the present method is obviously synthetically useful. The reaction mechanism appears to involve rare and interesting steps such as oxidative H-F elimination and is supported by several control experiments and DFT calculations. The results obtained will attract readers not only in organometallic and synthetic organic chemistry, but also in related fields.

The approach is reasonable. The text is written in a manner that is easily understood by a wide range of readers, and the quality of presentation is high. The quality of the data is satisfactory and the experimental procedures are detailed enough to be reproduced. The interpretation of the data and conclusions are considered reasonable and reliable. References are adequate. The abstract is clearly written, and the introduction and conclusions are clearly and appropriately stated. Overall, the paper is considered acceptable for publication in Nature Communications after some revisions. Questions and comments are listed below.

Response: We thank the Reviewer for the positive evaluation and strong support of this work.

1. Although the structure of DPEPhos is shown in Table 2, it should be moved to Table 1.

Response: We have made the revision according to the valuable suggestion.

2. In Table 1, entry 13, the reaction proceeds even in the absence of BQ, yielding the product in 45%. How can this result be explained? According to Figure 4, BQ is involved in the regeneration of LPd(OAc)₂.

Response: Due to a polar N–F bond, *N*-fluoro-sulfonamide **1** itself could serve as the terminal oxidant in the absence of BQ to facilitate the catalytic cycle of palladium catalyst. But nearly half of the substrate **1** would be consumed to amide by-product **3** (42% yield, entry 13 in Table 1). After screening of a number of external oxidants, BQ was found to be a suitable oxidant to effectively avoid the consumption of *N*-fluoro-sulfonamide **1**, while other oxidants, such as selectfluor, *m*CPBA, and LPO would suppress the desired dehydrogenation reaction (entries 14–16). What's more, when 0.2 equiv of BQ was added (entry 17), the yield of amide by-product **3** has decreased by 12% (from 42 to 30%), while the yield of desired product **2** has increased by 13% (from 45 to 58%), which further support that the use of BQ can avoid the consumption of *N*-fluoro-sulfonamide substrate and finally increase the yield of α,β -unsaturated imine **2**.

These information can be found in "Screening of the reaction conditions" part of the main text.

3. In Figure 4, the coordination bond of the alkene moiety to Pd in IV and VI should be drawn from the center of the alkene toward Pd.

Response: We have made the revision according to the valuable suggestion.

4. In Figure 3b, the authors show the effect of imine concentration on the product yield. I recommend that the authors perform this reaction using a syringe pump to slowly add the imine. This would keep the concentration of the imine low throughout the reaction. Under these conditions, the desired compound **5** may be obtained in a higher yield than those shown in Figure 3b.

Response: According to the suggestion of the Reviewer, we perform the reaction under standard conditions by slowly adding imine substrate **62** in dioxane (1.0 mL) using syringe pump for 15 min. To our delight, the yield of α,β -unsaturated imine **5** has indeed increased to 13%. But the dimer by-product **63** (25% yield, 50% yield based on imine **62**) still dominates the reaction outcome, suggesting a very fast dimerization process.

These information has been added in Figure 3b of the revised main text.

5. Have the authors attempted KIE (kinetic isotope effect) experiments using AcOD and/or compound D-4s? If the tautomerism from imine to enamine is the turnover-limiting step, as the authors have shown in Figure 4 based on the DFT calculations, KIE should be seen when the rates are compared between the reactions with AcOH or AcOD and compound 1 as a substrate, with AcOH and compound 4s (no D incorporated substrate of D-4s) or D-4s as a substrate, and with AcOH+4s or AcOD+D-4s. These data would further support the proposed mechanism.

Response: According to the suggestion of the Reviewer, kinetic isotope effect (KIE) studies have been conducted.

Performing intermolecular competition reaction with 0.5 equiv of **4s** and 0.5 equiv of **D-4s** under standard conditions for 5 min, the corresponding dehydrogenation products **4** and **D-4** are isolated in 25% overall yield with a ratio of 79:21 according to ^1H NMR ($P_{\text{H}}/P_{\text{D}} = 3.8$).

In addition, parallel reactions are performed under standard conditions at 40°C with either **4s** or **D-4s**. $k_{\text{H}}/k_{\text{D}} = 0.03/0.0104 = 2.9$

These results ($P_H/P_D = 3.8$, $k_H/k_D = 2.9$) indicate that the C–H bond-cleavage process may be involved in the rate-determining step, which is in accord with the DFT calculations in Figure 4.

This information has been added in "Mechanistic studies" part as Figure 3e of the revised main text, and the detailed procedures have been added in the revised Supplementary Information.

6. Can the authors reasonably explain why DPEPhos is more effective than other ligands?

Response: As an air-stable bidentate ligand, the existence of *ortho*-oxygen atom makes the arylphosphine in DPEPhos more electron-rich. This renders DPEPhos easily serving as a reductant to *in-situ* generate active Pd⁰ species (*Organometallics* **1995**, *14*, 1818-1826), and then initiate the dehydrogenation reaction (see Figure 4). Moreover, the electronically rich arylphosphine atoms benefit the coordination with palladium catalyst to form structurally stable and catalytically active species in dehydrogenation reaction. Both would make DPEPhos more effective compared to other ligands.

This information has been added in the secondary paragraph of "Theoretical calculations" part in the revised main text.

7. Have the authors attempted to observe LPd(H)(F) produced by the oxidative H-F elimination? The complex may be observable by ¹H and ¹⁹F NMR by a stoichiometric reaction at low temperature.

Response: Thanks to the Reviewer's valuable suggestion. For the convenience of NMR observation, a control experiment has been performed in NMR tube under standard conditions in acetone-*d*₆ at decreased temperature (40 °C). With stoichiometric palladium catalyst, there is too much metal salt in the tube to take NMR spectrum. Fortunately, in the presence of a catalytic

amount of palladium catalyst (10 mol%), the LPd(H)(F) complex has been successfully detected by both ^1H (δ -6.82 ppm) and ^{19}F (δ -186.9 ppm) NMR, which strongly support an oxidative H-F elimination process to form LPd(H)(F) complex.

These results have been added in Part 3.4 of the revised Supplementary Information, and also briefly mentioned at the end of "Theoretical calculations" part in the revised main text.

^1H NMR of LPd(H)(F) complex. The sample has been recorded in 500 MHz, acetone- d_6 at 25 °C

^{19}F NMR of LPd(H)(F) complex. The sample has been recorded in 565 MHz, acetone- d_6 at 25 °C

(Note: the unreacted *N*-fluoro-sulfonamide shows a peak at δ -51.3 ppm)

Response to the Reviewer 3's comments

Fu and co-workers describe a facile synthesis of α,β -unsaturated imines via palladium-catalyzed dehydrogenation. And this methodology has been applied to derivatization of biologically relevant molecules and a formal synthesis of (\pm)-alloyohimbane. However, this manuscript does not essentially solve the dehydrogenation of aliphatic imines, and it's just for the preparation of α,β -unsaturated imines from N-fluoroamide prepared with tedious multistep procedures. In fact, the challenges of preparation of α,β -unsaturated imines from imines were still not solved, such as easy hydrolysis and preferential dimerization. In addition, the direct α,β -dehydrogenation of imines have already been reported more than a decade ago, and it subsequently serves as an intermediate to take place other transformations (*Adv. Synth. Catal.* 2009, 351, 1229-1232; *Chem. Asian J.* 2009, 4, 1712-1716; *Angew. Chem. Int. Ed.* 2011, 50, 3920-3924). From the perspective of publication, I would recommend publication after address the following points.

Response: We thank the Reviewer for the positive evaluation and the support of this work.

1) The authors elaborate the developments in α,β -dehydrogenation of carbonyl compounds in the introduction in detail, and related references have been cited. However, relevant references have been missed, such as *J. Org. Chem.* 2019, 84, 8267-8274; *Org. Lett.* 2021, 23, 1611-1615; *Org. Lett.* 2023, 25, 4429-4433; *Org. Biomol. Chem.* 2017, 15, 7317-7320; *Angew. Chem. Int. Ed.* 2018, 57, 16205-16209; *Adv. Synth. Catal.* 2018, 360, 4774-4783. And they should be cited under the corresponding topics.

Response: Thanks to the Reviewer's valuable suggestion.

References *Org. Biomol. Chem.* 2017, 15, 7317-7320; *Adv. Synth. Catal.* 2018, 360, 4774-4783; *Org. Lett.* 2021, 23, 1611-1615; and *Org. Lett.* 2023, 25, 4429-4433 have been cited as ref. 17, 18, 23, and 24, respectively, under the topic of aldehydes/ketone dehydrogenation.

References *Angew. Chem. Int. Ed.* 2018, 57, 16205-16209 and *J. Org. Chem.* 2019, 84, 8267-8274 have been cited as ref. 36, and 38, respectively, under the topic of amide dehydrogenation.

2) As the authors state, aliphatic imines are easy hydrolysis. But various α,β -unsaturated imines are isolated via a flash column chromatography in this manuscript. Precautions need to be listed in the Supplementary Information. The yields range from decent to moderate, but there is no discussion on the reasons for not having "100% yield" in these reactions. What are the side products? Hydrolysis products? Sequential dehydrogenative products (γ,δ -dehydrogenative products)?

Response: Due to the conjugated stabilization, α,β -unsaturated imines are more stable compared to their parent aliphatic imines. They could be isolated by flash column chromatography, preferably within one hour. A precaution "It would be better to finish the purification within one hour to avoid the product decomposition" has been added in "Methods" part of the revised main text and the revised Supplementary Information.

In addition to the desired α,β -unsaturated imines, defluorinated aliphatic amides (such as **3**) are the main by-products, together with some unidentified hydrolysis by-products. These information has been added in the first paragraph of "Substrate scope" part of the revised main text.

3) Regarding the substrate scope, except for benzene sulfonyl groups on nitrogen, are other

substituents feasible? Such as phenyl, alkyl? In the introduction, the author stated the oxidative dehydrogenation of N-heterocycles provides an efficient access to functionalized aromatic N-heterocycles. But dehydrogenation of N-heterocycles hadn't been exhibited in this manuscript. Maybe benzene sulfonyl groups on nitrogen are crucial. However, what are the advantages of this method compared with condensation between aldehydes and *p*-toluenesulfonamide? And the preparation of raw materials is complicated for this method. And a gram-scale should be presented to prove the practicability of the method rather than 0.20 mmol scale.

Response: For phenyl or alkyl-substituted amines and *N*-heterocycles, unfortunately, it's impossible to obtain their corresponding *N*-fluoroamine substrates due to the high polarity of N–F bond. Despite the construction of α,β -unsaturated imines mainly relies on condensation of ene-aldehydes and *p*-toluenesulfonamide in the current stage, tedious procedures are usually required for the preparation of functionalized ene-aldehydes. What's more, for the transformation of some complex alkaloids or nitrogen-containing compounds to the corresponding α,β -unsaturated imines, it would first need an oxidative conversion of the nitrogen-containing moiety to aldehyde that is not easily realized and controlled, followed by dehydrogenation of the resultant aldehyde to ene-aldehyde and a final condensation with *p*-toluenesulfonamide. This nitrogen atom "cut-install" reaction sequence shows no advantage in step- and atom-economies. Thus, our developed aliphatic amide dehydrogenation provides an important supplementary method for the synthesis of α,β -unsaturated imines, especially for the late-stage functionalization of complex nitrogen-containing compounds.

A large-scale synthesis (5.0 mmol) has existed in Part 3.2 of Supplementary Information, providing the desired dehydrogenation product **2** in 74% yield. This data has been added in Table 1 as note [d] of the revised main text.

4) About reaction mechanism, a KIE experiment should be implemented to confirm the rate-determining step. The author stated a low concentration of the imine intermediate benefitted the avoidance of its dimerization (Figure 3b). However, only two sets of data are given. To better illustrate the tendency, more data should be given. Apart from DFT calculations, is there any other evidence about a concerted five-membered-ring (Pd-F-N-C-H) transition state TSI rather than the stepwise oxidative addition followed by β -H elimination, such as kinetic study?

Response: According to the suggestion of this Reviewer as well as Reviewer 2, kinetic isotope effect (KIE) studies with **4s** and **D-4s** have been conducted. The results ($P_H/P_D = 3.8$, $k_H/k_D = 2.9$) indicate that the C–H bond-cleavage process may be involved in the rate-determining step, which is in accord with the DFT calculations in Figure 4.

These information has been added in "Mechanistic studies" part as Figure 3e of the revised main text, and the detailed procedures have been added in the revised Supplementary Information.

In Figure 3b, in addition to the current two sets of data, more experiments in low concentration have been performed according to the suggestion of the Reviewer. However, further diluting the reaction mixture would give inferior yields of the desired product **5** (15% in 0.01 M, 9% in 0.005 M), probably due to a slow dehydrogenation process under low concentration of palladium catalyst and BQ.

These information has been added in "Mechanistic studies" part of the revised main text.

We have tried some kinetic studies according to the suggestion of the Reviewer. But the obtained data are inconsistent probably due to a fast dehydrogenation process within 15 min. Fortunately, conducting a control experiment in NMR tube under standard conditions in acetone-*d*₆ at 40 °C, the LPd(H)(F) complex has been successfully detected by both ¹H (δ -6.82 ppm) and ¹⁹F (δ -186.9 ppm) NMR, which support an oxidative H-F elimination process to form LPd(H)(F) complex.

These results have been added in Part 3.4 of the revised Supplementary Information, and also briefly mentioned at the end of "Theoretical calculations" part in the revised main text.

¹H NMR of LPd(H)(F) complex. The sample has been recorded in 500 MHz, acetone-*d*₆ at 25 °C

^{19}F NMR of LPd(H)(F) complex. The sample has been recorded in 565 MHz, acetone- d_6 at 25 °C
 (Note: the unreacted *N*-fluoro-sulfonamide shows a peak at δ -51.3 ppm)

5) Minor points: All “ α,β ” in the manuscript should be italicized. The line 282, “reveale” to “reveal”.

Response: We have made the revision according to the valuable suggestion.

REVIEWERS' COMMENTS

Reviewer #2 (Remarks to the Author):

The authors have addressed most of the reviewers' comments seriously, and I believe that the manuscript is ready for publication in Nature Communications.

One comment: in the responses to the reviewers 2 and 3's comments about the observation of LPd(H)(F) by NMR, the authors mentioned that 'These results have been added in Part 3.4 of the revised Supplementary Information, and also briefly mentioned at the end of "Theoretical calculations" part in the revised main text.', but I do not seem to find such a description that should be included at the end of "Theoretical calculations" part in the revised main text.

Reviewer #3 (Remarks to the Author):

The authors have revised the manuscript and ESI in this revised manuscript. Many errors have been corrected, and the comments and questions have been addressed. However, based on manuscript, the reaction is not actually the α,β -dehydrogenation of aliphatic imines, but an pre-fluorination of amides, followed by a dehydrogenation for the preparation of α,β -unsaturated imines. Therefore, the direct α,β -desaturation of aliphatic imines have not been solved yet. This manuscript may mislead the reader that the α,β -desaturation of aliphatic imines has been solved. In that sense, the manuscript about the α,β -desaturation of aliphatic imines is inappropriate to a certain extent. On this basis, I suggest deleting the expressions about the α,β -desaturation of aliphatic imines.

Response to the Reviewer 2's comments

Reviewer #2 (Remarks to the Author):

The authors have addressed most of the reviewers' comments seriously, and I believe that the manuscript is ready for publication in Nature Communications.

One comment: in the responses to the reviewers 2 and 3's comments about the observation of LPd(H)(F) by NMR, the authors mentioned that 'These results have been added in Part 3.4 of the revised Supplementary Information, and also briefly mentioned at the end of "Theoretical calculations" part in the revised main text.', but I do not seem to find such a description that should be included at the end of "Theoretical calculations" part in the revised main text.

Response: We thank the Reviewer for the positive evaluation and strong support of this work.

A description "The LPd^{II}(H)(F) complex has been successfully detected by both ¹H and ¹⁹F NMR (for details, see Part 3.4 in the Supplementary Information)" has been added in the second paragraph of "Theoretical calculations" part in the revised main text.

Response to the Reviewer 3's comments

Reviewer #3 (Remarks to the Author):

The authors have revised the manuscript and ESI in this revised manuscript. Many errors have been corrected, and the comments and questions have been addressed. However, based on manuscript, the reaction is not actually the α,β -dehydrogenation of aliphatic imines, but a pre-fluorination of amides, followed by a dehydrogenation for the preparation of α,β -unsaturated imines. Therefore, the direct α,β -desaturation of aliphatic imines has not been solved yet. This manuscript may mislead the reader that the α,β -desaturation of aliphatic imines has been solved. In that sense, the manuscript about the α,β -desaturation of aliphatic imines is inappropriate to a certain extent. On this basis, I suggest deleting the expressions about the α,β -desaturation of aliphatic imines.

Response: We thank the Reviewer for the positive evaluation and strong support of this work.

According to the suggestion of the reviewer, a description "The reaction proceeds via an initial Pd(0)-mediated oxidative H-F elimination and a subsequent Pd(II)-catalyzed desaturation process with smooth potential energy surface. The *in-situ* generation of aliphatic imine intermediate avoids the necessity of purification and prevents the accumulation of imine intermediate for further dimerization" has been revised in the last paragraph of "Introduction" part in the main text, which gives a brief explanation on the success of dehydrogenation to α,β -unsaturated imines.